# Interaction of mental comorbidity and physical multimorbidity predicts length-of-stay in medical inpatients

**Sophia Stahl-Toyota**[1]*, **Christoph Nikendei**[1], **Ede Nagy**[1], **Stefan Bönsel**[2],
**Ivo Rollmann**[1], **Inga Unger**[3], **Julia Szendrödi**[4], **Norbert Frey**[5], **Patrick Michl**[6],
**Carsten Müller-Tidow**[7], **Dirk Jäger**[8], **Hans-Christoph Friederich**[1], **Achim Hochlehnert**[2]

1 Department of General Internal Medicine and Psychosomatics, Medical University Hospital, Heidelberg, Germany, 2 Department of Medicine Controlling, Medical University Hospital, Heidelberg, Germany, 3 Nursing Management, Department of Internal Medicine, Medical University Hospital, Heidelberg, Germany, 4 Department of Endocrinology and Clinical Chemistry, Medical University Hospital, Heidelberg, Germany, 5 Department of Cardiology, Angiology and Pneumology, Medical University Hospital, Heidelberg, Germany, 6 Department of Gastroenterology, Hepatology and Infectious Diseases, Medical University Hospital, Heidelberg, Germany, 7 Department of Hematology, Oncology and Rheumatology, Medical University Hospital, Heidelberg, Germany, 8 Department of Medical Oncology, Medical University Hospital, National Center for Tumor Diseases, Heidelberg, Germany

* sophia.stahl-toyota@med.uni-heidelberg.de

**Data Availability Statement:** Minimal data for this study cannot be shared publicly because of identifying personal patient information gathered in clinical routine that underlies personal data

## Abstract

### Background

Mental comorbidities of physically ill patients lead to higher morbidity, mortality, health-care utilization and costs.

### Objective

The aim of the study was to investigate the impact of mental comorbidity and physical multi-morbidity on the length-of-stay in medical inpatients at a maximum-care university hospital.

### Design

The study follows a retrospective, quantitative cross-sectional analysis approach to investigate mental comorbidity and physical multimorbidity in internal medicine patients.

### Patients

The study comprised a total of n = 28.553 inpatients treated in 2017, 2018 and 2019 at a German Medical University Hospital.

### Main measures

Inpatients with a mental comorbidity showed a median length-of-stay of eight days that was two days longer compared to inpatients without a mental comorbidity. Neurotic and somato-form disorders (ICD-10 F4), behavioral syndromes (F5) and organic disorders (F0) were leading with respect to length-of-stay, followed by affective disorders (F3), schizophrenia

protection regulations imposed by Ethikkommission Medizinische Fakultät Heidelberg. Data will be made available upon request from Ethikkommission Medizinische Fakultät Heidelberg via email (ethikkommission-I@med.uni-heidelberg. de) for researchers who meet the criteria for access to confidential data.

**Funding:** The authors received no specific funding for this work.

**Competing interests:** The authors have declared that no competing interests exist.

and delusional disorders (F2), and substance use (F1), all above the sample mean length-of-stay. The impact of mental comorbidity on length-of-stay was greatest for middle-aged patients. Mental comorbidity and Elixhauser score as a measure for physical multimorbidity showed a significant interaction effect indicating that the impact of mental comorbidity on length-of-stay was greater in patients with higher Elixhauser scores.

## Conclusions

The findings provide new insights in medical inpatients how mental comorbidity and physical multimorbidity interact with respect to length-of-stay. Mental comorbidity had a large effect on length-of-stay, especially in patients with high levels of physical multimorbidity. Thus, there is an urgent need for new service models to especially care for multimorbid inpatients with mental comorbidity.

## Introduction

The analysis of the influence of mental comorbidity and physical multimorbidity in hospitalized patients on health-economic parameters is of great interest in health care systems [1].

The course and prognoses of physical diseases is determined decisively by the concurrent presence of mental comorbidities, such as depression or anxiety disorder [2, 3]. In addition, physical diseases are often accompanied by a pronounced psychosocial strain [4]. Regarding socio-economic parameters, it is well known that mental comorbidities of patients with physical diseases lead to prolonged length-of-stay in hospitals [5–7], higher morbidity and mortality rates [2–5], as well as lower quality of life [6]. It is therefore an imperative that comorbid mental disorders are diagnosed in somatic hospitals and treated in a timely manner. Cardiovascular patients with mental comorbidity, for example, show a significant increase of the average length-of-stay from 8.9 (± 0.3) to 13.2 (± 0.7) days [8]. The physical-mental interplay deteriorates dramatically the physical health condition associated with significant higher hospital costs. However, regarding the insurance payment systems, extra resources engaged in such cases are not adequately represented [8–10].

Besides potential mental comorbidities, the presence of additional physical conditions in somatically ill patients also shows a negative influence on the length-of-stay at the hospital [11, 12]. For in-patients the prevalence of a multimorbidity status, defined as the presence of at least two chronic conditions by the World-Health-Organisation, is about 80% [13, 14]. There do exist further definitions of multimorbidity, that are relevant to address the present research question. Aubert *et al.* have investigated eight different definitions for multimorbidity, which all had a moderate to medium separative power in their ability to predict 30-day-hospital-readmission and prolonged length-of-stay [15]. Among others, they defined multimorbidity using the Elixhauser-van Walraven comorbidity index and the absolute number of conditions. Mueller *et al.* described an increased length-of-stay of 4.7 (± 10.7) days for multimorbid cases in comparison to non multimorbid cases [14].

Taken together, these findings emphasize that the incorporation of physical multimorbidity and mental comorbidity is important in models of socio-economic target parameters such as length-of-stay. A recent review on length-of-stay prediction lists only few studies that actually take comorbidities into account [16]. And those current studies that do discuss length-of-stay in the context of multimorbidity, should be interpreted with caution due to potential

contortion of results due to the fact that physical multimorbidity and mental comorbidity might interact with each other [1]. Thus the relationship between mental comorbidity and length-of-stay might be changing depending on the level of physical multimorbidity. This is of great importance for a more realistic and valid prediction of the influence of mental comorbidity on the length-of-stay in medical inpatients. In addition, a limiting factor can be seen in the restricted sample sizes [8].

Therefore, the aim of the presented study was to investigate the influence of (1) mental comorbidity, (2) physical multimorbidity and (3) the interaction of physical multimorbidity and mental comorbidity on the length-of-stay using a large database of a center for internal medicine at a German university hospital. The University clinic of Heidelberg belongs to the publicly funded hospitals, which is the group, next to charitable and private hospitals, that provides nearly 50% of all the available beds in Germany [17]. University hospitals treat around 2 million patients annually in a stationary setting [18]. Another way to classify hospitals is by four levels of care, ranging from basic to regular, specialized and maximum-care. University clinics usually cover all medical disciplines and are classified as maximum-care. Concerning mental comorbidities in medically ill patients, the "consultation-liasion services" by psychiatric and psychosomatic specialists is the model offered most commonly to patients in the medical departments throughout Germany [19]. According to the typology by Kathol *et al.*, the cohort described here was treated on a Type II medical consultation-liaison unit for low mental and medium to high medical acuity [20].

The analyses presented here may help to allocate resources more adequately at medical hospitals and to provide early psychosocial interventions for patients with additional needs. Moreover, it could be imagined that a future predictive model estimates the expected length-of-stay at the beginning of a hospital stay and supports clinical decisions for early interventions, such as a proactive psychosomatic and psychiatric consultation service. Since resources are limited, targeted interventions should be possible that are of economic and medical importance by early identification of especially critical cases.

## Materials and methods

### Study design and participants

The study comprised a retrospective data analysis of all inpatient cases of the years 2017, 2018 and 2019 at the Center of Internal Medicine of the University Hospital in Heidelberg, Germany. After data preprocessing and removal of one patient for whom the gender was unknown, N = 28,553 cases had no missing values for the variables of interest and met the inclusion criteria, which included cases of 20,193 patients who (1) were at least 18 years old, (2) were admitted to one of the internal medicine units, (3) had no main diagnosis of the ICD-10 (International Statistical Classification Of Diseases And Related Health Problems, 10th revision, German Modification) code chapter V for psychiatric diseases, (4) and had a length-of-stay of at least two days. These criteria were selected so that only stationary patients who stayed at least one night and who were admitted with a primarily somatic main diagnosis were included. The source of the data were the medical records that entailed use of resources via diagnostic and therapeutic effort, so only diagnostic ICD-10 codes that were relevant to the respective hospital stay were available. Data preprocessing involved merging cases that were direct follow-up admissions. Hospital stays that were medically associated, e.g. due to complications, and occurred within a 30 days interval were thus counted as a single case.

## Measures

**Outcome length-of-stay.** The outcome variable of interest was the length-of-stay. It was extracted from clinical routine data directly, representing the number of days within one case that the patient stayed at the hospital.

**Physical multimorbidity.** For representing physical multimorbidity, the Elixhauser score was retrieved via the R package *comorbidity scores* for all comorbidity diagnoses, not including the main diagnosis [21]. As Bartlett *et al.* [22] did, the Elixhauser score was modified to exclude the four groups related to mental diseases (psychoses, depression, drug and alcohol abuse) to separate physical from mental comorbidities, leaving 27 of the 31 groups represented by the score.

**Mental comorbidity.** We defined mental comorbidity according to Wolff *et al.* [23] as any secondary diagnosis code from Chapter V (F0-F9) of the ICD-10. Several previous studies omitted codes representing diseases such as dementia, delirium or nicotine abuse [8, 11, 23, 24]. However, since there was no consensus on the reasons for omitting particular diagnosis codes, all codes were kept in the definition for this analysis. The mental comorbidity count was the number of ICD-10 codes that fulfilled the above definition for mental comorbidity.

**Other variables.** As potential confounders, the patients' gender, age at hospitalization and main diagnosis ICD-10 chapter were included in the analysis.

**Statistical analyses.** To describe the characteristics of the study population, we report the mean, standard deviation, median, and range for continuous variables. For categorical variables, the absolute number and the ratio of each value is presented. All characteristics were computed for the total population as well as for two groups separated by presence or absence of mental comorbidity. To evaluate the difference of the variable distributions between these groups, the Pearson's chi-squared test was computed for categorical variables and the Mann-Whitney-U test for numerical variables. In addition, each variable's correlation with the outcome parameter length-of-stay was computed as either the robust correlation coefficient with percentage bend (r) or correlation ratio (η) for exploratory data analysis to aid in the model selection process [25].

For the analysis of length-of-stay differentiated by mental comorbidity count, F-category of mental comorbidity, age groups with and without mental comorbidity, and increasing Elixhauser score, the mean and 95% confidence interval in the respective subgroups were computed and plotted. To further assess the relationship between the explanatory variables with length-of-stay, we realized single variate as well as multivariate analysis to analyse main and interaction effects with negative binomial regression. Since this type of model has a parameter to control for overdispersion, it is well-suited for the characteristic long-tailed distribution of length-of-stay [26, 27]. As in those studies, the analysis was performed on the basis of hospitalizations, so that one patient may be represented by several cases. Model performance was evaluated mainly based on the Akaike Information Criterion (AIC), Mean Absolute Error (MAE) and Root Mean Squared Error (RMSE) [28, 29]. The derivation of the model with R code is portrayed in S1 File. The choice of methods put a high focus on intuitive interpretability, as reliable reasoning is essential for trusting the results [16].

The data preparation and characteristics description was implemented in Python 3.11 (scipy 1.8.0 [30], pingouin 0.5.0 for statistical tests [31]). All remaining steps were computed with R 4.1.2 (comorbidity 1.0.0 [21] for Elixhauser score determination, mass 7.3.54 [32] for negative binomial regression).

### Ethical approval

This study was approved by the Ethics committee of the Medical faculty of the University of Heidelberg (No. S-690/2021). As only clinical routine data were used, the need for consent was waived by the Ethics committee.

## Results

### Characteristics of study participants

The characteristics of the 28.553 cases that met the inclusion criteria are displayed in Table 1. Out of all cases, 15.2% were diagnosed with at least one mental comorbidity. These patients with mental comorbidity are younger and have significantly higher Elixhauser scores. The mean and median length-of-stay in the different main diagnosis ICD10 chapter groups are included in S1 Table. The correlation ratio for the main diagnosis chapter and length-of-stay was $\eta = 0.19$, and for gender $\eta = 0.002$.

### Length-of-stay with vs. without mental comorbidity

The most frequent mental diagnoses belonged to the categories F0 organic diseases (n = 1394), F1 substance use (n = 1319), F4 neurotic, stress and somatoform disorders (n = 1042) and F3 affective disorders (n = 879). The number of cases in the other F-categories were less than 200 cases per category.

Mean length-of-stay of patients without a mental comorbidity was 8.8 days (95% confidence interval, CI95 = [8.7, 8.9]; median = 6), while mean length-of-stay of patients with one or more mental comorbidities was 15.2 days (CI95 = [14.4, 15.9]; median = 8). Thus, on average patients with mental comorbidities stayed 6.4 days longer. The difference of the median was 2 days. The common language effect size, for which deviation from 0.5 expresses significance, was f = 0.4.

**Table 1. Characteristics of the sample of internal medicine inpatients with and without mental comorbidity.**

| Variable | Measure | Total | Without mental comorbidity | With mental comorbity |
|---|---|---|---|---|
| Cases | N (%) | 28553 (100%) | 24225 (85%) | 4328 (15%) |
| Gender | female (%) | 11334 (39.7%) | 9540 (39.4%) | 1794 (41.5%) |
| Age at hospitalization (y) | mean [SD] | 64.0 [16.9] | 64.2 [16.9] | 62.9 [17.2] |
| | median | 66 | 67 | 64 |
| | min-max | 18–108 | 18–108 | 18–98 |
| Physical comorbidities Elixhauser score | mean [SD] | 2.8 [2.0] | 2.7 [2.0] | 3.4 [2.2] |
| | median | 3 | 2 | 3 |
| | min-max | 0–15 | 0–13 | 0–15 |
| Mental comorbidity count | mean [SD] | 0.2 [0.5] | 0.0 [0.0] | 1.3 [0.7] |
| | median | 0 | 0 | 1 |
| | min-max | 0–7 | 0–0 | 1–7 |
| **Length-of-stay (d)** | mean [SD] | 9.8 [13.6] | **8.8** [9.8] | **15.2** [25.3] |
| | median | 6 | **6** | **8** |
| | min-max | 2–463 | 2–274 | 2–463 |
| | 90th percentile | 20 | 19 | 32 |

SD: standard deviation, CI95: 95% confidence interval, 90th percentile with interpolation = 'nearest'.

## Length-of-stay by number of mental comorbidities

Fig 1 shows a more detailed view of the relationship between length-of-stay and number of mental comorbidities (correlation coefficent r = 0.09, CI95 = [0.08, 0.1]). With increasing number of mental comorbidities, the average length-of-stay increased. Out of all n = 4328 cases with mental comorbidities, the largest group (n = 3411) with one mental comorbidity diagnosis had an average length-of-stay of 13.8 days, which is 5.0 days longer than cases without any mental comorbidity and without overlapping confidence intervals.

## Length of stay differentiated for the F-category of mental comorbidity

The length-of-stay of patients with mental comoribities is further differentiated by the particular type of mental comorbidity in Fig 2. Patients with neurotic disorders (F4) had the highest mean length-of-stay with 23.01 days. Patients with behavioral syndromes (F5) stayed on average 19.28 days, but the group was small with 78 cases and had high variance, followed by organic disorders (F0) with 18.94 days and affective disorders (F3) with 14.97 days on average. Patients with substance use (F1) had rather short length-of-stay with on average 10.93 days. This was slightly above the sample mean of 9.8 days.

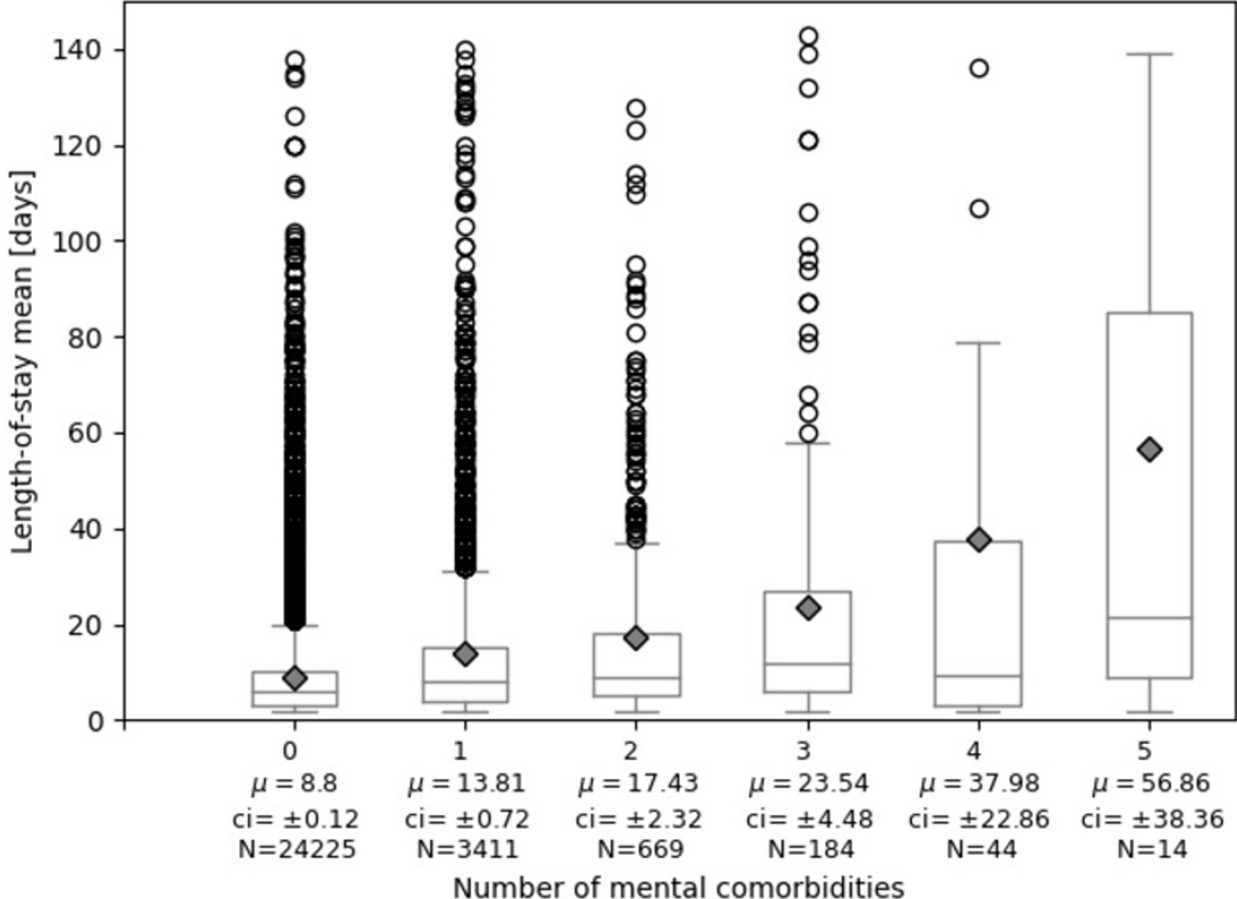

**Fig 1. Length-of-stay in relation to number of mental comorbidities.** μ: mean length-of-stay, plotted as diamonds; ci: 95% confidence interval; N: number of cases. The number of mental comorbidities on the x-axis is displayed up to 5, as the number of cases with 6 or 7 mental comorbidities is less than 10. The y-axis is cut-off at 140 days for better discernability of the boxes. S1 Fig displays the plot with all outliers.

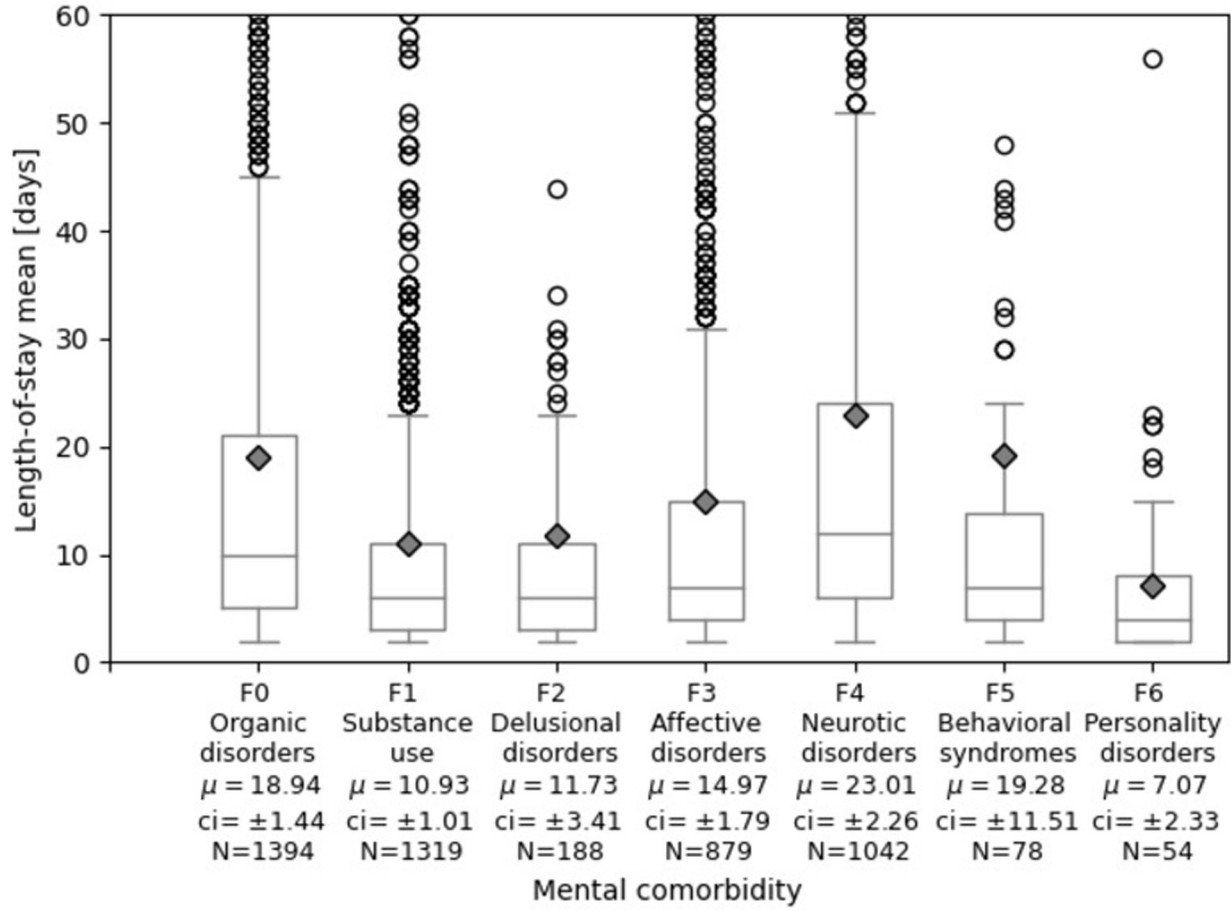

**Fig 2. Length-of-stay differentiated by mental comorbidity spectrum.** μ: mean length-of-stay, plotted as diamonds; ci: 95% confidence interval; N: number of cases that have a diagnosis in the respective F-category (ICD-10 range starting with the characters F0-F6). Cases that have mental comorbidity diagnoses in several F-categories are counted in each one separately and are therefore represented in multiple boxes. The y-axis is cut-off at 60 days for better discernability of the boxes. S2 Fig displays the plot with all outliers.

## Length-of-stay for different age groups with and without mental comorbidity

Fig 3 displays the mean length-of-stay for different age groups with and without mental comorbidity, S2 Table shows the corresponding numerical values. There was a negative correlation (r = -0.05, CI95 = [-0.06, -0.04]) of age at hospitalization with length-of-stay. The ratio of patients with mental comorbidity was highest in the age groups that encompass 40 to 59 years at hospitalization (18%). Length-of-stay without mental comorbidity was around the population mean (9.8 days) for all age groups (group mean ranging from 7.03 to 10.19 days). Mean length-of-stay for cases with mental comorbidity ranged from 7.58 to 20 days and was higher within each age group than for those without mental comorbidity. Mean length-of-stay was highest for middle-aged groups with mental comorbidity: patients in their thirties (mean 20 days, 11 days longer than without mental comorbidity), were followed by those in their fifties (mean 19.12, 9 days longer than without mental comorbidity).

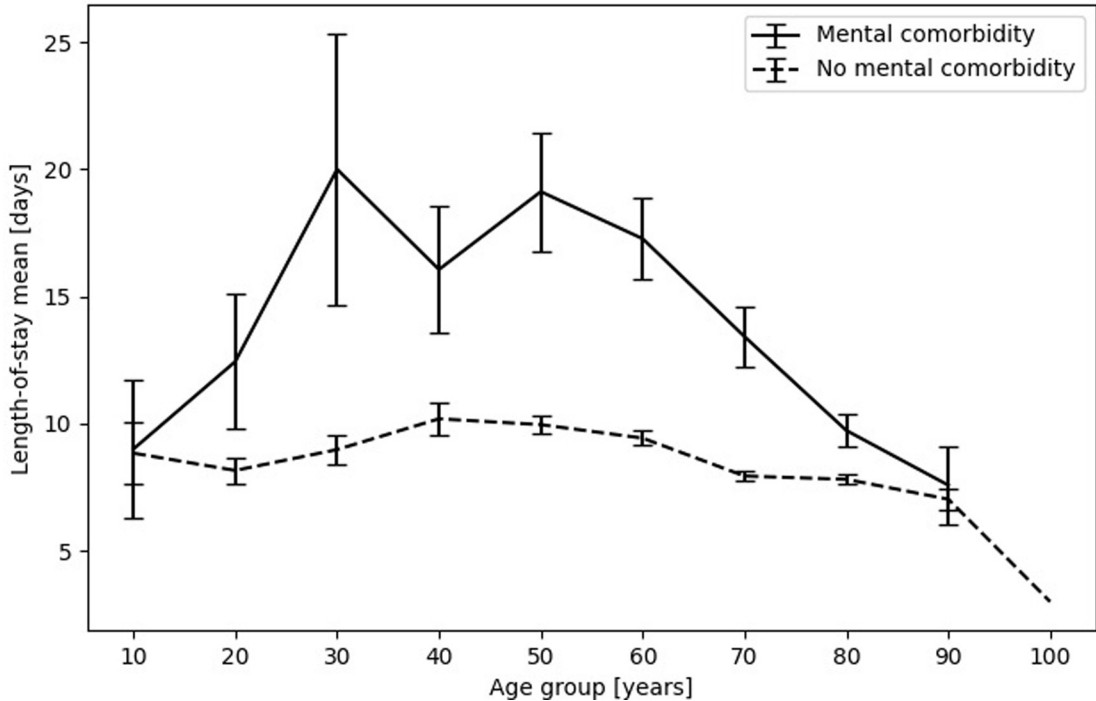

**Fig 3. Length-of-stay differentiated by age group and presence of mental comorbidity.** Age group naming: 10 encompasses age at hospitalization 18–19 years, 20 encompasses 20–29 years, 30 eoncompasses 30–39 years etc. The numerical values matching this figure are in Table 2. Error bars indicate the 95% confidence interval.

### Length-of-stay in relation to physical multimorbidity with and without mental comorbidity

Fig 4 shows the relationship between physical multimorbidty, represented by the Elixhauser score, and length-of-stay separately for cases with and without mental comorbidty; S3 Table shows the corresponding numerical values. The overall correlation coefficient for the variable Elixhauser score was r = 0.17 (CI95 = [0.16, 0.18]). Increasing Elixhauser score without mental comorbidity was associated with moderate increase in length-of-stay. Additional mental comorbidity was associated with stronger increases in length-of-stay dependent on the level of physical multimorbidity. The lines of mental comorbidity and Elixhauser Score in Fig 4 cross at about Elixhauser score 1 indicating that mental comorbidity has an increasingly additional effect on length-of-stay starting from an Elixhauser score above 1. This relationship indicates an interaction between mental comorbidity and physical multimorbidity and was therefore included as an interaction term in the regression model.

### Negative binomial regression model with Elixhauser score and mental comorbidity presence

Table 2 displays the results of negative binomial regression analysis for length-of-stay. According to multivariate analysis, each additional year of age at hospitalization reduced the average length-of-stay by 0.4%. Presence of mental comorbidity increased average length-of-stay by 13.1%. An increase in Elixhauser score (which may cover many comorbidity codes, so not directly comparable to mental comorbidity presence) increased average length-of-stay by 13.8%. In interaction with a mental comorbidity, it increased by an additional 8.5%. This interaction between presence of mental comorbidity and Elixhauser score was significant. S2 File is a case

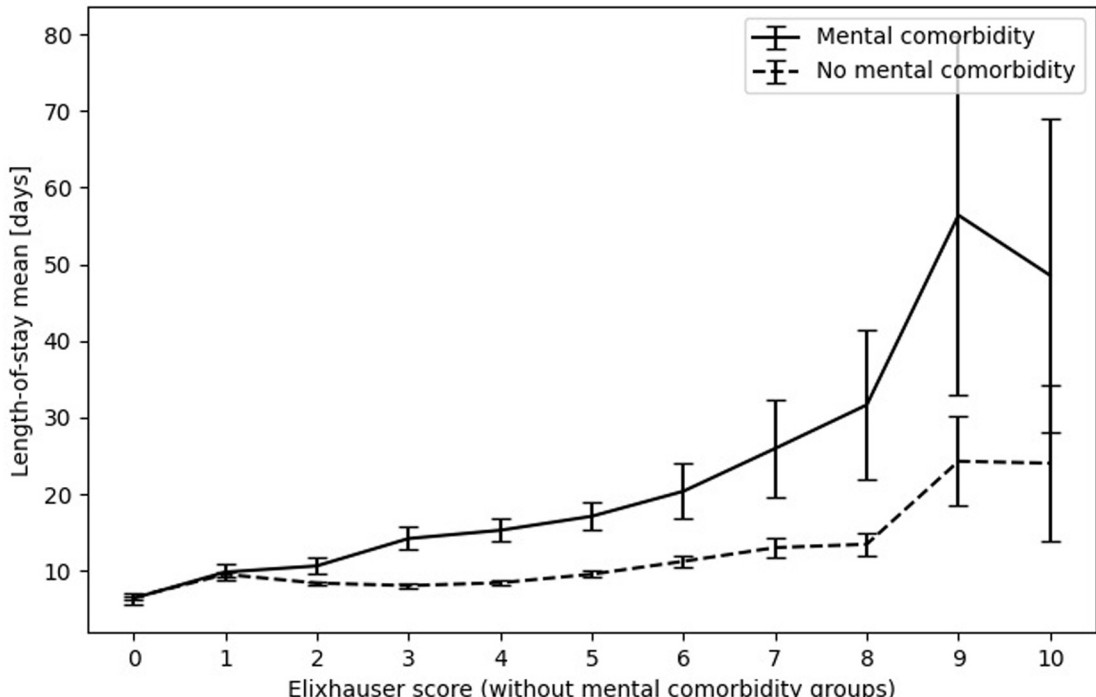

**Fig 4. Length-of-stay for increasing Elixhauser score with and without mental comorbidity.** The Elixhauser score on the x-axis is displayed up to 10, as the number of cases with higher scores were ≤ 20. The numerical values matching this figure are shown in S3 Table. Error bars indicate the 95% confidence interval.

simulation tool where individual predictions can be simulated and the contribution of each variable to the prediction can be retraced according to the model's coefficients as derived in S3 File.

## Discussion

The present study underlines the importance of mental comorbidity on health-economic outcomes such as length-of-stay. Besides organic diseases (F0), substance use (F1), neurotic, stress and somatoform disorders (F4) and affective disorders (F3) were the most frequent comorbid mental diagnoses at a maximum-care hospital of internal medicine. The greatest length-of-stay in the present sample was observed for middle-aged medical inpatients (30–59 years) and patients with neurotic, stress and somatoform disorders (F4). The main finding of the present study is that the influence of mental comorbidity on the length-of-stay interacts with the level of physical multimorbidity.

We found that the mean length-of-stay of patients with mental comorbidity was 6.4 days longer than the length-of-stay of patients without mental comorbidities. The median differed by 2 days. These findings match a review by Jansen *et al.*, who reviewed studies that calculated the difference in length-of-stay with and without mental comorbidity [1]. Across 20 studies, the mean difference between controls and the group with mental comorbidities was on average 8.9 days (SD = 13.6). The median of the mean differences was 5 days. The individual studies showed great heterogeneity. The median among the mean length-of-stay reported by the 20 reviewed studies was 13.9 days with and 9.2 days without mental comorbidity, which is close to the mean length-of-stay of 15.2 and 8.8 days found in our study for the two groups, respectively. Another study by Beeler *et al.* showed that an additional non-depression diagnosis was independently associated with an increased length-of-stay by 10% and an ancillary depression

**Table 2. Negative binomial regression analysis of length-of-stay.**

| | Univariate analysis | | | Multivariate analysis | | |
|---|---|---|---|---|---|---|
| Variable | IRR | CI95 | p-value | IRR | CI95 | p-value |
| **Gender** | | | | | | |
| Male | / | / | / | / | / | / |
| Female | 1.005 | 0.985–1.026 | 0.62 | 1.019 | 1–1.038 | 0.05 |
| **Age at hospitalization** | 0.994 | 0.994–0.995 | $< = 0.001$*** | 0.996 | 0.995–0.997 | $< = 0.001$*** |
| **Mental comorbidity present** | 1.722 | 1.676–1.77 | $< = 0.001$*** | 1.131 | 1.081–1.183 | $< = 0.001$*** |
| **Somatic comorbidities Elixhauser score** | 1.11 | 1.105–1.116 | $< = 0.001$*** | 1.138 | 1.131–1.144 | $< = 0.001$*** |
| **Main Diagnosis ICD-10 chapter** | | | | | | |
| I Infectious and parasitic diseases | / | / | / | / | / | / |
| II Neoplasms | 1.377 | 1.311–1.446 | $< = 0.001$*** | 1.486 | 1.42–1.554 | $< = 0.001$*** |
| XIV Genitourinary system | 1.179 | 1.1–1.264 | $< = 0.001$*** | 1.132 | 1.063–1.207 | $< = 0.001$*** |
| XIX Pregnancy, childbirth and puerperium | 1.218 | 1.132–1.31 | $< = 0.001$*** | 1.053 | 0.985–1.126 | 0.13 |
| VI Nervous system | 1.081 | 0.902–1.305 | 0.41 | 1.014 | 0.859–1.204 | 0.87 |
| XIII Musculoskeletal and connective tissue | 0.886 | 0.803–0.978 | $< = 0.05$* | 0.993 | 0.907–1.088 | 0.88 |
| XI Digestive system | 0.946 | 0.899–0.995 | $< = 0.05$* | 0.905 | 0.864–0.947 | $< = 0.001$*** |
| IV Endocrine, nutritional and metabolic dis. | 0.879 | 0.827–0.934 | $< = 0.001$*** | 0.888 | 0.84–0.939 | $< = 0.001$*** |
| VII Eye and adnexa | 0.822 | 0.47–1.558 | 0.52 | 0.858 | 0.512–1.53 | 0.58 |
| III Blood and immune mechanisms | 0.852 | 0.763–0.953 | $< = 0.01$** | 0.856 | 0.773–0.949 | $< = 0.01$** |
| XII Skin and subcutaneous tissue | 0.825 | 0.651–1.059 | 0.12 | 0.825 | 0.662–1.038 | 0.09 |
| X Respiratory system | 0.797 | 0.748–0.849 | $< = 0.001$*** | 0.78 | 0.735–0.826 | $< = 0.001$*** |
| XV Origin in perinatal period | 0.769 | 0.543–1.121 | 0.15 | 0.702 | 0.507–0.991 | $< = 0.05$* |
| VIII Ear and mastoid process | 0.574 | 0.331–1.062 | 0.06 | 0.68 | 0.406–1.193 | 0.15 |
| IX Circulatory system | 0.748 | 0.715–0.783 | $< = 0.001$*** | 0.633 | 0.606–0.66 | $< = 0.001$*** |
| XVII Findings not elsewhere classified | 0.633 | 0.515–0.785 | $< = 0.001$*** | 0.59 | 0.487–0.72 | $< = 0.001$*** |
| XVIII Injury, poisoning | 0.578 | 0.534–0.626 | $< = 0.001$*** | 0.588 | 0.547–0.634 | $< = 0.001$*** |
| XXI Factors influencing health status | 0.412 | 0.37–0.458 | $< = 0.001$*** | 0.391 | 0.354–0.432 | $< = 0.001$*** |
| **Interaction of Mental comorbidity present and Somatic comorbidities Elixhauser score** | | | | 1.085 | 1.073–1.097 | $< = 0.001$*** |

IRR: incidence rate ratio (represents change in length-of-stay in terms of percentage, as determined by distance from 1, per unit increase for continuous variables and per presence of category for categorical variables); CI95: 95% confidence interval. Note that several main diagnosis groups encompass less than 100 cases, refer to S2 Table for exact case counts.

even by 24% [33]. The analysis, however, excluded several mental comorbidities and the prevalence of depression was 4.9% [33].

These previous studies have not considered the physical-mental interplay with respect to length-of-stay, which have led to distorted results. For an average medical inpatient with an Elixhauser score of 2, our model associated the addition of mental comorbidity with a 3.0 days extension length-of-stay. When the Elixhauser score was 7, the length-of-stay difference between a patient with compared to without mental comorbidity was 17.2 days (see S2 File for interactive individual case simulations and the derivation of these numbers). Thus, our study provides evidence from a large sample of inpatients at a maximum-care university hospital that mental comorbidity predicts length-of-stay dependent on physical multimorbidity.

There was a tendency that younger patients had greater length-of-stay, with each additional year reducing the length-of-stay by 0.04% in the multivariate analysis, which is in accordance with previous studies [26, 27]. It is notable that the gap with respect to length-of-stay between the group with mental comorbidity in comparison to those without mental comorbidity is highest in middle-aged inpatients (30–59 years). This finding may also be characteristic for

patients at a maximum-care hospital compared to a basic and regular care hospital with less elderly and frail patients suffering from organic diseases (F0). However, the findings also underline the impact of mental comorbidity on length-of-stay in patients younger than 60 years. These patients show greater psychosocial vulnerability to physical morbidity, as the consequences of illness appear in critical phases of career planing, starting a family and partnership. Higher levels of suffering in 30–59 year old patients may exceed own ressources resulting in a cascade of dysfunctions that contribute to a greater length-of-stay.

Regarding the mental comorbidity subcategories, we would have expected that especially organic disorders (F0) such as dementia and delirium in addition to physical comorbidities would be the main comorbidity-related factors increasing length-of-stay. Our study, however, showed that cases with neurotic, stress-related and somatoform disorders (F4) had the highest length-of-stay. This ICD-10-chapter, however, includes several anxiety disorders like phobic, panic, and obsessive-compulsive disorders as well as disease processing and somatoform disorders that are a domain for psychotherapy.

The fact that different kinds of depression, which are summarized in ICD-10-chapter F3, are associated with a longer length-of-stay, as reported previously [33, 34], is corroborated by the present study, although the impact on length-of-stay was lower compared to mental diseases from ICD-10 chapters F0, F4, F5.

The present findings have several clincial implications. First of all, there is a great need to develop new concepts of integrated care for medical inpatients with mental comorbidity and asscociated complex care needs. Mental comorbidity has a prominent effect on the length-of-stay, especially in multimorbid inpatients. The interaction between mental comorbidity presence and Elixhauser score can be interpreted in the sense that for patients with mental comorbidity, the effect of increasing levels of multimorbidity on length-of-stay was stronger. This pertains only to the subgroup of patients with mental comorbidity, which in this cohort was 15%. The true ratio, however, is expected to be higher if systematic screening for mental comorbidities would be performed. Therefore, it is necessary to diagnose and treat mental comorbidity especially in physically multimorbid patients upon admission to hospital and along inpatient treatment. Knowing which comorbidities have the strongest effect on length-of-stay can help to allocate resources to address the specific needs of these patients early on.

Consultation-liasion services have the limitation that they traditionally become active only on request. Given the prevalence of mental comorbidity of about 35% in medical inpatients [6], the present findings of 15% show that physicans in internal medicine identify less than half of the patients with mental comorbidity and even less are seen by the psychosomatic or psychiatric consultation-liasion services. To address these limitations, for inpatients with mental comorbidity, a systematic screening for mental comorbidity of all patients should be implemented. Furthermore, a proactive consultation is necessary, so that more patients receive the complex care they need. Alternatively, the psychosomatic or psychiatrist should be part of the ward team. This means he or she is actively involved in patients' ongoing inpatient care in the sense of a liaison service model. The latter expands and improves clincial care to a bio-psycho-social care model. This suggestion is similar to a behavioral intervention team that was shown to be a promising way of decreasing length-of-stay in general medical units [35] as well as the Proactive Integrated Consultation-Liaison Psychiatry (PICLP) for which effectiveness is currently under evaluation [36].

As for most studies on length-of-stay prediction [16], a limitation is that the generalizability to other settings may be questionable. The sample was limited to the Center of Internal Medicine, but more applicable conclusions could be drawn if data for the entire hospital or even other hospitals were included. S1 Table shows that the entire ICD-10 spectrum of internal

disases was covered by the cohort, however, the cases might not be representative for all hospitals in Germany, as our data represents a maximum-care university hospital.

Even though the variables employed in this study, age, gender, primary diagnosis and comorbidity, are among the key features that appear in multiple length-of-stay studies listed by a recent review [16], information of potential confounders such as disease severity as well as social and economic status of the patients was missing.

Furthermore, the prevalence of mental comorbidity was lower than the previously reported 35% for a smaller cohort at the same hospital [6], indicating that approximately half of the mental comorbidities may not have been identified by the specialists in internal medicine. This may have influenced the findings as certain diagnoses were more easily detected by physicians.

In order to reach more clinically relevant conclusions, the model would also need to differentiate different subcategories of physical comorbidities and incorporate the different subcategories of mental comorbidities as their severity is not equal and impact on length-of-stay can be presumed to vary according to the descriptive results presented here.

All relevant diagnoses for the hospital stay counted equally towards the analysis, not distinguishing between diagnoses placed at the beginning or end of the stay, allowing no differentiation between preexisting or a newly developed mental comorbidity. In addition, diagnoses from the medical history of the patient were not available, as only those codes that were available in the medical records of the current hospital stay were included in this data set.

Although directly connected cases were merged and the remaining multiple hospitalizations for the same patient were due to independent admission reasons, the observations are not entirely independent of each other, since the same patients were evaluated more than once. Even though this kind of analysis on a hospitalization-level is consistent with similar studies [26, 27], future work should take the patient-level information into account to address this potential bias. Also a multi-level model with the hospital departments as clusters may be considered for model improvement, as their mean lengths-of-stay varied (S4 Table).

Besides improving the model by adapting the choice of features, the model building and evaluation process itself could also be performed with less risk of overfitting. Instead of one single iteration that uses the entire dataset for model building, methods such as cross-validation for model tuning and splitting the available data into training, validation and test sets could be employed [16].

## Conclusions

In conclusion, this study confirmed that the length-of-stay of patients with a mental comorbidity is longer than of patients without such a comorbidity at a German university hospital.

With detailed differentiation among subgroups divided by age, mental comorbidity subcategories and physical multimorbidity, we were able to highlight those subgroups with especially high length-of-stay. Furthermore, the difference between length-of-stay in medical inpatients with and without mental comorbidity increases with the level of physical multimorbidity.

In the future, the context of these individual variables should be analyzed even further in multivariate predictive models that allow more precise prediction of increased hospital length-of-stay in order to deliver actionable insights regarding hospital resource allocation.

## Supporting information

**S1 Table. Length-of-stay (LOS, days) mean and median per main diagnosis chapter.** -P: without psychiatric comorbidity. +P: with psychiatric comorbidity. Sorted by LOS mean difference between cases with and without psychiatric comorbidity. Chapters with total number

of cases less than 500 grouped into Z Other: ("III", "VI", "VII", "VIII", "XII", "XIII", "XV", "XVII", "XXI"). Test type U: Mann-Whitney-U. f: common language effect size (value of 0.5 means no significant difference, deviation from 0.5 expresses siginifance), p-value significance: *p ≤ 0.05, **p≤ 0.01, ***p≤ 0.001.
(DOCX)

**S2 Table. Length-of-stay by age group and presence of mental comorbidity.** These are the underlying numbers for Fig 3. N: number of cases; LOS: length-of-stay; CI95: 95% confidence interval.
(DOCX)

**S3 Table. Length-of-stay for increasing Elixhauser score with and without mental comorbidity.** These are the underlying numbers for Fig 4. N: number of cases; LOS: length-of-stay; CI95: 95% confidence interval.
(DOCX)

**S4 Table. Hospital departments.** Number of cases ratio and mean length-of-stay with and without mental comorbidity. -P: without psychiatric comorbidity. +P: with psychiatric comorbidity.
(XLSX)

**S1 File. Model selection.** R code and output as well as comments for model selection process.
(PDF)

**S2 File. Case simulation.** In the sheet "case_simulation", enter values of interest in columns C-G to see the predicted length-of-stay in column H. The details of the contribution of each variable to the prediction are in columns M-S. The other sheets contain the coefficients that are used for the calculation. Highlighted yellow are the two numbers for example predictions mentioned in the discussion.
(XLSX)

**S3 File. Derivation of case simulation calculations.**
(DOCX)

**S1 Fig. Length-of-stay in relation to number of mental comorbidities with outliers.** μ: mean length-of-stay, plotted as diamonds; ci: 95% confidence interval, also shown by error bars; N: number of cases. The number of mental comorbidities on the x-axis is displayed up to 5, as the number of cases with 6 or 7 mental comorbidities is less than 10.
(TIF)

**S2 Fig. Length-of-stay differentiated by mental comorbidity spectrum with outliers.** μ: mean length-of-stay, plotted as diamonds; ci: 95% confidence interval, also shown by error bars; N: number of cases that have a diagnosis in the respective F-category (ICD-10 range starting with the characters F0-F6). Cases that have mental comorbidity diagnoses in several F-categories are counted in each one separately and are therefore represented in multiple barsboxes.
(TIF)

## Author Contributions

**Conceptualization:** Christoph Nikendei, Ivo Rollmann, Hans-Christoph Friederich, Achim Hochlehnert.

**Data curation:** Sophia Stahl-Toyota, Stefan Bönsel, Achim Hochlehnert.

**Formal analysis:** Sophia Stahl-Toyota, Ede Nagy, Ivo Rollmann.

**Funding acquisition:** Hans-Christoph Friederich.

**Investigation:** Sophia Stahl-Toyota, Stefan Bönsel, Achim Hochlehnert.

**Methodology:** Sophia Stahl-Toyota, Ede Nagy, Ivo Rollmann.

**Project administration:** Christoph Nikendei.

**Resources:** Christoph Nikendei, Stefan Bönsel, Inga Unger, Julia Szendrödi, Norbert Frey, Patrick Michl, Carsten Müller-Tidow, Dirk Jäger, Hans-Christoph Friederich.

**Supervision:** Christoph Nikendei, Hans-Christoph Friederich.

**Validation:** Ede Nagy.

**Visualization:** Sophia Stahl-Toyota.

**Writing – original draft:** Sophia Stahl-Toyota, Achim Hochlehnert.

**Writing – review & editing:** Christoph Nikendei, Ede Nagy, Stefan Bönsel, Ivo Rollmann, Inga Unger, Julia Szendrödi, Norbert Frey, Patrick Michl, Carsten Müller-Tidow, Dirk Jäger, Hans-Christoph Friederich.

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
