## [Decision Letter · Decision Letter 0]

13 Mar 2023

PONE-D-23-03643­Interaction of mental comorbidity and physical multimorbidity predicts length-of-stay in medical inpatientsPLOS ONE

Dear Dr. Stahl-Toyota,

Thank you for submitting your manuscript to PLOS ONE. After careful consideration, we feel that it has merit but does not fully meet PLOS ONE’s publication criteria as it currently stands. Therefore, we invite you to submit a revised version of the manuscript that addresses the points raised during the review process.

We look forward to receiving your revised manuscript.

Kind regards,

Sebastien Kenmoe

Academic Editor

PLOS ONE

Reviewers' comments:

Reviewer's Responses to Questions

**Comments to the Author**

1. Is the manuscript technically sound, and do the data support the conclusions?

Reviewer #1: Partly

Reviewer #2: Yes

Reviewer #3: Partly

2. Has the statistical analysis been performed appropriately and rigorously? 

Reviewer #1: No

Reviewer #2: No

Reviewer #3: No

3. Have the authors made all data underlying the findings in their manuscript fully available?

Reviewer #1: No

Reviewer #2: No

Reviewer #3: No

4. Is the manuscript presented in an intelligible fashion and written in standard English?

Reviewer #1: Yes

Reviewer #2: Yes

Reviewer #3: Yes

5. Review Comments to the Author

Reviewer #1: Interaction of mental comorbidity and physical multimorbidity predicts length-of-stay in medical inpatients

This observational study explores the impact of mental comorbidity and physical multimorbidity on the length-of-stay (LoS) in medical inpatients. Hospital LoS is an important outcome in the planning of healthcare services and control of medical costs. The authors focused on the effects of mental and physical multimorbidity instead of age, gender and primary diagnosis as the ‘usual suspects’ in studies that aim to predict LoS. Like most studies in this field, a limitation of this paper is that data collection and modelling are restricted to one hospital: the internal medicine department of the University Hospital in Heidelberg, Germany. However, there are other limitations and methodological issues.

1. STROBE-guidelines state that “readers need information on setting and locations to assess the context and generalizability of a study’s results”. Information is lacking on special features of the university hospital in the region and the healthcare system in Germany in general. The authors conclude that “there is a great need to develop new concepts of integrated care for medical inpatients with mental comorbidity”, but they are not clear about existing concepts other then consultation-liasion services. The typology of Kathol et al. (1992) could be helpful to clarify the regional and national context.

2. The STROBE-statement also points at the importance of clearly defining all variables and reporting missing values for each variable of interest and for each step in the analysis. However, in this study “no missing values” was one of the inclusion criteria. Other criteria are not clarified: why were cases with less than two hospital days and with main diagnosis of the ICD-10 code chapter V for psychiatric diseases excluded? In the discussion section, the authors mention that for patients 60+ years old early transfer in geriatric or general hospitals occurs regularly. Should the analyses not have been restricted to the age group 18-60? Organic diseases (n=1394), substance use (n=1319), neurotic and somatoform disorders (n=1042) and affective disorders (n=879) were the most frequent mental diagnoses, but these are not main diagnoses of the ICD-10 code chapter V?

Also, information is lacking on how diagnoses were recorded. The limitations section makes clear that no differentiation could be made between preexisting or newly developed mental comorbidity. But in 911 cases two up to five diagnoses were registered, including (unlike other studies) dementia, delirium or nicotine abuse. In the final analysis, only “Mental comorbidity present” is used, which in terms of severity and impact puts very different mental diagnoses on an equal footing. Sensitivity analyses could help to clarify the effects of this approach.

3. In total, N=28,553 cases met the inclusion criteria of which 8.360 admissions (29% of all cases) were follow-up admissions. The authors acknowledge that this is a violation of the assumption in regression analysis that data should be independent, but make no effort to explore this possible source of bias. It could have been helpful to compare characteristics in Table 1 of patients with one or multiple admissions (although single admissions in the beginning of 2017 and at the end of 2019 could be part of multiple admissions that fall outside the observation period).

4. In the statistical analyses section, the authors state that robust correlation coefficients with percentage bend (r) or correlation ratios (η) were calculated, but it is not clear what these un-directional correlation coefficients would add to the univariate and multivariate analyses regressing LoS on mental comorbidity and physical multimorbidity.

Gender, age and main diagnosis ICD-10 are seen as potential confounders, but main diagnosis is not included in the negative binomial regression model. Supplementary material 1 shows large LoS differences between cases with and without psychiatric comorbidity per ICD-10 chapter, which suggest relevant co-occurrence of main diagnosis and prevalence of mental problems: injury and poisoning (suicide risk?) or neoplasms (depression and anxiety?). Moreover, the statistical analysis plan does not include model comparison, sensitivity analyses and methods to check model fit.

5. Table 1 in this paper reports characteristics of the sample of internal medicine inpatients and tests differences between cases with and without mental comorbidity. However, STROBE-guidelines state that inferential measures and significance tests should be avoided in descriptive tables.

6. Table 2 presents the main analyses in terms of incidence rate ratios. The model includes an interaction-effect of mental comorbidity and physical multimorbidity, which concerns the primary hypothesis. Yet in the text (line 268) this is based on figure 4 (which shows different LoS-values per Elixhauser score for patients with and without mental comorbidity), but no interaction-effect of mental comorbidity and age although figure 3 shows different LoS-values per age-group for patients with and without mental comorbidity. And probably there is also an interaction effect of age and Elixhauser score. Therefor it is unclear how model selection came about and how models were compared. The fit of the final model or predictive power is not discussed.

In the discussion section the authors calculated some LoS-estimates based on the model (3 days extension for someone with mental comorbidity and an Elixhauser score of 2 and 17.2 days for an Elixhauser score of 7), but these calculations are difficult to follow and are presented as LoS-values instead of expected averages with confidence intervals or estimates with prediction intervals. The actual number of predicted hospital days can be calculated from the unexponentiated model coefficients, but these values are not reported.

7. The authors conclude that “increasing levels of multimorbidity are associated with a growing positive influence of mental comorbidity on the length-of-stay.” But this is not what the interaction-term implies. The 8.5% increase per additional physical morbidity concerns only patients with mental comorbidity. For this group the effect of physical multi morbidities increases somewhat more compared to patients without mental comorbidity. In the discussion section the authors fail to give a substantive interpretation of this complex interaction effect. This interpretation should support the conclusion that new concepts of integrated care need to be developed.

8. In the discussion section, the authors point at the limitations of this study. The recent review of studies on the prediction of hospital length of stay by Stone et al. (2022) could be a useful reference in this regard.

References

R.G. Kathol, H.H. Harsch, R.C.W. Hall, et al. Categorization of types of medical/psychiatry units based on level of acuity. Psychosomatics, 33 (1992), pp. 376-386

Stone K, Zwiggelaar R, Jones P, Mac Parthaláin N (2022) A systematic review of the prediction of hospital length of stay: Towards a unified framework. PLOS Digit Health 1(4): e0000017. https://doi.org/10.1371/journal.pdig.0000017

Reviewer #2: The authors aimed to investigate the impact of concurrent mental comorbidity and physical multimorbidity on the length of stay in medical inpatients at a maximum-care university hospital. This cross-sectional study has 28,553 inpatients treated in Germany between 2017 and 2019. The authors identified mental health conditions using the ICD-10 chapter and physical comorbidity using the Elixhauser score. Although the research presents an important issue, the authors need to address some issues.

Reviewer #3: Dear Authors,

You investigate the influence of comorbid mental disorders on the length of inpatient treatment in somatic hospitals. In a retrospective design,routine clinical data was evaluated and a correlation between comorbidity and length of stay was found. The conclusion from from the results is that new care models are needed for multimorbid patients with concomitant mental illness.

The question of the present study is undoubtedly of great relevance and the the analysis is based on an impressive number of cases collected across several units. The manuscript is also clearly and comprehensibly written.

However, some questions or points of discussion arise for me.

(1) It seems to me that the literature in the introduction is not up to date. In addition, it partly refers to other health systems that may only be comparable to a limited extent.

(2) How exactly were the mental comorbidities diagnosed? The diagnosis of mental illness is often quite difficult. If the diagnosis is not made by specialists, misdiagnoses cannot be ruled out.

(3) The statistical analysis seems to me to have room for improvement. Since the data were collected across different departments with probably different mean lengths of stay, I think a multi-level model with the departments as clusters could be considered. If varying slopes were allowed, the effects could be estimated in a more differentiated way.

(4) It would be helpful for the reader to have more information on the departments involved. Could the lengths of stay and the proportions of cases with comorbidities be broken down by department?

Overall, a publication of the study is very desirable, but I would advise a revision beforehand.

Yours sincerely

6. PLOS authors have the option to publish the peer review history of their article (what does this mean?). If published, this will include your full peer review and any attached files.

Reviewer #1: **Yes: **A.I. Wierdsma

Reviewer #2: No

Reviewer #3: No

---

## [Author Response · Author response to Decision Letter 0]

12 May 2023

Please refer to the file Response_to_Reviewers for a formatted version of this text:

Journal Requirements

Thank you for giving us the chance to specify this important aspect. The need for consent was waived by the ethics committee. We added this information in the Methods section:

‘This study was approved by the Ethics committee of the Medical faculty of the University of Heidelberg (No. S-690/2021). As only clinical routine data were used, the need for consent was waived by the ethics committee.’

Thank you for this comment and request. However, the de-identified data set still includes the variables gender, age, main diagnosis and comorbidities. Together with the information in which time frame and at which hospital the patients were treated, the data set was deemed as potentially sensitive by the ethics committee. We agree with the suggestion for the Data Availability statement: "Minimal data for this study cannot be shared publicly because of identifying personal patient information gathered in clinical routine that underlies personal data protection regulations imposed by Ethikkommission Medizinische Fakultät Heidelberg. Data will be made available upon request from Ethikkommission Medizinische Fakultät Heidelberg via email (ethikkommission-I@med.uni-heidelberg.de) for researchers who meet the criteria for access to confidential data."

Reviewer 1:

Dear Reviewer #1,

Thank you very much for reviewing this manuscript. We have considered each of your important indications thoroughly. 

Comments:

1) STROBE-guidelines state that “readers need information on setting and locations to assess the context and generalizability of a study’s results”. Information is lacking on special features of the university hospital in the region and the healthcare system in Germany in general. The authors conclude that “there is a great need to develop new concepts of integrated care for medical inpatients with mental comorbidity”, but they are not clear about existing concepts other then consultation-liasion services. The typology of Kathol et al. (1992) could be helpful to clarify the regional and national context.

Thank you for setting the perspective of the international readership into focus. We provided more information about the German health care system in the introduction. According to the typology of Kathol et al., the type of care has been stated more precisely to improve the understanding of the setting.

2) The STROBE-statement also points at the importance of clearly defining all variables and reporting missing values for each variable of interest and for each step in the analysis. However, in this study “no missing values” was one of the inclusion criteria. Other criteria are not clarified: why were cases with less than two hospital days and with main diagnosis of the ICD-10 code chapter V for psychiatric diseases excluded? In the discussion section, the authors mention that for patients 60+ years old early transfer in geriatric or general hospitals occurs regularly. Should the analyses not have been restricted to the age group 18-60? Organic diseases (n=1394), substance use (n=1319), neurotic and somatoform disorders (n=1042) and affective disorders (n=879) were the most frequent mental diagnoses, but these are not main diagnoses of the ICD-10 code chapter V?

Also, information is lacking on how diagnoses were recorded. The limitations section makes clear that no differentiation could be made between preexisting or newly developed mental comorbidity. But in 911 cases two up to five diagnoses were registered, including (unlike other studies) dementia, delirium or nicotine abuse. In the final analysis, only “Mental comorbidity present” is used, which in terms of severity and impact puts very different mental diagnoses on an equal footing. Sensitivity analyses could help to clarify the effects of this approach.

Thank you very much for your feedback. We agree that the definition of the cohort needs more detailed clarification and would like to address your suggestions in these ways:

a) Missing values: Indeed, the data completeness was high in this data set and only one patient had a missing value for gender. Please refer to the R code in S4 Supporting Information for the check for missing values. We reworded the passage about inclusion criteria to state more clearly that there were indeed no otherwise missing values, and we did not remove any further data for that reason. 

b) Cases with only one hospital day were excluded: Cases with a value for length-of-stay less than two days meant that they were treated as ambulant cases and not as stationary, as was the cohort of interest for this study. We added this clarification to the inclusion criteria reasoning.

c) Chapter V for psychiatric diseases excluded for main diagnosis: This study was not about psychiatric patients with additional mental comorbidities. They are taken care of in a psychiatric department with a different reimbursement system. Our study is about somatic patients with mental comorbidity for whom the potential need for psychiatric support is hypothesized to be underestimated. We also added this reason to the inclusion criteria. 

d) Transfer of patients 60+ years old: The sentence in the discussion seems to be misleading as the number of patients who are transferred is not as high as was suggested here. The hospital is a maximum care hospital, so it is also representative for patients aged 60+. The sentence was removed from the discussion.

e) Chapter V for psychiatric diseases included for comorbidity diagnoses: The raw data set had two different variables for the main diagnosis and comorbidities. Both variables contain ICD-10 codes. Therefore, we could use the main diagnosis variable to filter out mental main diagnoses, while determining mental comorbidities for all remaining main diagnoses from the comorbidity variable. 

f) Information on how diagnoses were recorded: Only diagnoses were recorded that have entailed use of resources / medication / diagnostic and therapeutic effort, so diagnoses that were relevant for the focus of treatment for the respective hospital stay. This information was added after the inclusion criteria and to a note regarding this limitation in the discussion.

g) Severity of different mental diagnoses: Of course, the impact of schizophrenia is probably more severe on length-of-stay than an addiction to tobacco, for example. We are aware that the model we present puts all mental diseases on the same level, and future models should account for the differences in severity. We added this thought explicitly to the limitations and would declare the presented model as a kind of benchmark model for which intuitive interpretation was the main aim. 

3) In total, N=28,553 cases met the inclusion criteria of which 8.360 admissions (29% of all cases) were follow-up admissions. The authors acknowledge that this is a violation of the assumption in regression analysis that data should be independent, but make no effort to explore this possible source of bias. It could have been helpful to compare characteristics in Table 1 of patients with one or multiple admissions (although single admissions in the beginning of 2017 and at the end of 2019 could be part of multiple admissions that fall outside the observation period).

In the study design and the discussion, we added a note to distinguish between follow-up admissions that were already merged during data pre-processing and truly different cases. We absolutely agree that the topic of readmissions needs to be addressed. This is an important aspect which we plan to address in dedicated future studies as a prediction target variable in the sense of readmission within a particular time frame but consider it out of scope for the present study.

4) In the statistical analyses section, the authors state that robust correlation coefficients with percentage bend (r) or correlation ratios (η) were calculated, but it is not clear what these un-directional correlation coefficients would add to the univariate and multivariate analyses regressing LoS on mental comorbidity and physical multimorbidity.

Gender, age and main diagnosis ICD-10 are seen as potential confounders, but main diagnosis is not included in the negative binomial regression model. Supplementary material 1 shows large LoS differences between cases with and without psychiatric comorbidity per ICD-10 chapter, which suggest relevant co-occurrence of main diagnosis and prevalence of mental problems: injury and poisoning (suicide risk?) or neoplasms (depression and anxiety?). Moreover, the statistical analysis plan does not include model comparison, sensitivity analyses and methods to check model fit.

Thank you for requesting more details on the model selection process, as of course, more effort than presented initially took place. We show the derivation of the model with the corresponding R code and output in the new S4 Supporting Information where we also portray the measures used for checking model fit (AIC, RMSE, MAE). 

The main diagnosis was included in the negative binomial regression model, but the coefficients had been missing in the table and were now added. 

The correlation coefficients and correlation ratios were calculated as part of the exploratory analysis to determine which variables to include in the model. We also assume that it is advantageous to present these values for which the interpretation should be familiar to the readership.

5) Table 1 in this paper reports characteristics of the sample of internal medicine inpatients and tests differences between cases with and without mental comorbidity. However, STROBE-guidelines state that inferential measures and significance tests should be avoided in descriptive tables.

We removed the respective column from the table and transferred the most relevant test statistic to the paragraph about length-of-stay with vs. without mental comorbidity.

6) Table 2 presents the main analyses in terms of incidence rate ratios. The model includes an interaction-effect of mental comorbidity and physical multimorbidity, which concerns the primary hypothesis. Yet in the text (line 268) this is based on figure 4 (which shows different LoS-values per Elixhauser score for patients with and without mental comorbidity), but no interaction-effect of mental comorbidity and age although figure 3 shows different LoS-values per age-group for patients with and without mental comorbidity. And probably there is also an interaction effect of age and Elixhauser score. Therefor it is unclear how model selection came about and how models were compared. The fit of the final model or predictive power is not discussed.

In the discussion section the authors calculated some LoS-estimates based on the model (3 days extension for someone with mental comorbidity and an Elixhauser score of 2 and 17.2 days for an Elixhauser score of 7), but these calculations are difficult to follow and are presented as LoS-values instead of expected averages with confidence intervals or estimates with prediction intervals. The actual number of predicted hospital days can be calculated from the unexponentiated model coefficients, but these values are not reported.

We are glad to provide more details regarding the model selection, please refer to S4 Supporting Information. The effect of interaction between mental comorbidity and age is explored there, as well. The model improvement was not as strong as the effect of mental comorbidity and Elixhauser, however. A potential interaction between age and Elixhauser may be incorporated in future models that aim to optimize model prediction. Here, we wanted to focus on the effect of mental comorbidity and therefore tried to minimize model complexity. 

As interpretability was among our highest priorities, we had prepared a case simulation excel file to perform those calculations. Since these calculations appear to be of interest to the reader, we naturally can include them as S5 Supporting Information. We also added a note in the discussion regarding the usage of this file. 

7) The authors conclude that “increasing levels of multimorbidity are associated with a growing positive influence of mental comorbidity on the length-of-stay.” But this is not what the interaction-term implies. The 8.5% increase per additional physical morbidity concerns only patients with mental comorbidity. For this group the effect of physical multi morbidities increases somewhat more compared to patients without mental comorbidity. In the discussion section the authors fail to give a substantive interpretation of this complex interaction effect. This interpretation should support the conclusion that new concepts of integrated care need to be developed.

Thank you for pointing out the need for clearer communication of the interaction implications. We reworded the quoted sentence to stress the fact that the interaction is only relevant for the sub-group with mental comorbidity and moved it to the paragraph about the clinical implications.

8) In the discussion section, the authors point at the limitations of this study. The recent review of studies on the prediction of hospital length of stay by Stone et al. (2022) could be a useful reference in this regard.

That is a very valuable resource. We have incorporated a few of the shortcomings of current LOS prediction research listed by the authors that also apply to our study.

Reviewer 2:

Dear Reviewer #2,

Thank you very much for reviewing our manuscript. Your advice was very helpful and pointed out important issues of our study. 

Comments:

1) Introduction: The authors wrote, “Moreover, it could be imagined that a future predictive model estimates the expected length-of-stay at the beginning of a hospital stay and supports clinical decisions for early interventions, such as a proactive psychosomatic and psychiatric consultation service.”, I understand this idea is attractive and agree that it is crucial to allocate resources appropriately. However, in practice, principal and additional diagnoses are only recorded after the care is complete, i.e. at the end, so I’m unsure how the model suggested by the authors can be fitted.

It is true that the final entry of diagnoses is required by law for reimbursement purposes only at the end of the hospital stay when care is completed. In practice, however, diagnoses are often recorded as soon as they are identified. Most diagnoses are identified at admission. If further diagnoses are placed, they are available as soon as the diagnosis is made. The challenge for a would be to make the diagnoses available to the model as soon as they are identified.

2) Method: The authors should provide more information on the health care system in Germany and, if they need more space, shorten the results section as people from different jurisdictions will need help understanding the system in Germany.

Thank you for setting the perspective of the international readership into focus. We provided more information about the German health care system in the introduction. The setting of the psychiatric/medical unit has also been stated more precisely.

3) Method: Physical multimorbidity -> need more details on which diagnosis (primary, additional, or all) was used to determine this. I’m guessing it’s all diagnoses, but I need clarification.

Only the additional diagnoses were used to determine the Elixhauser score, the primary diagnosis was not included. We added this indeed important detail to the respective section in the methods.

4) Method: Mental comorbidity -> Authors should consider not including F7 in the mental comorbidity as it is an intellectual disability, which is not considered a mental disorder. Generally, people with intellectual disability are known to have a longer stay in the hospital.

It is true that an intellectual disability is not to be set equal to a mental disorder. In order to consider not including F7, we checked how many cases would be affected by this change. 36 cases have an F7 comorbidity, so the expected effect on the result of the present study’s results would be negligible. Therefore, we would stick to the current data pre-processing decisions, also as other similar studies list very heterogeneous filtering criteria for mental disorders. In future work we would definitely reconsider this decision, as more detailed analysis of the mental disorder subcategories is desired anyhow. 

5) Statistical analysis: The authors mentioned that the analysis was performed at the hospitalisation level. Can you please clarify whether you can identify patients from your data? While I agree that the analysis should be performed at the hospitalisation level, I want to know how the authors control for within-patient variation. Records from the same patients could share some characteristics and have a similar outcome; not accounting for it could produce biased results. Also, the authors’ statistical methods assumed data independence to be valid.

Yes, we can identify patients from our data. In the study design and the discussion, we added a note to distinguish between follow-up admissions that were already merged during data pre-processing and truly different cases. This merge of case records should prevent bias through complications from the same patients with the same admission reason resulting in different case numbers in the raw data. Only age and gender are expected to be shared among the remaining different cases of the same patients, but main diagnosis, Elixhauser score and mental comorbidity presence can vary. In the future, however, we plan to examine the topic of readmission more closely, as information such as readmission within a particular timeframe would also be an interesting target variable.

6) Results: Can you please add the 90th percentile for the length of stay in Table 1? The data is skew, so the 90th percentile will help the reader understand the distribution better.

To emphasize the overdispersion, that is typical for length-of-stay distributions, is a very good idea; an additional row was added at the bottom of Table 1 for the 90th percentile which was derived with the interpolation option set to ‘nearest’.

7) Results: Figures 1 & 2: I believe the box plot with 6 boxes for each mental health group will be more informative than the current graphs.

Thank you for this good idea, we have replaced the figures with box plots and included the measures plotted before in the x axis labels. 

Reviewer 3:

Dear Reviewer #3,

Thank you very much for reviewing our manuscript. Your comments were very valuable and helped to improve the description of our study. 

Comments:

1) It seems to me that the literature in the introduction is not up to date. In addition, it partly refers to other health systems that may only be comparable to a limited extent.

Thank you for taking the international perspective of the readership into focus. We added a short description of the hospital landscape in Germany and how the presented cohort is classified. We also added Freitas et al. (12) and Stone et al. (16) to the literature.

2) How exactly were the mental comorbidities diagnosed? The diagnosis of mental illness is often quite difficult. If the diagnosis is not made by specialists, misdiagnoses cannot be ruled out.

We agree that the diagnosis of mental illness should be performed by specialists. That is why most mental comorbidities that are reported here are diagnosed by a psychiatric specialist who is called in by the treating physician if a mental comorbidity is suspected for a patient. Misdiagnoses of course cannot be ruled out in principle. But the risk for this cohort is reduced due to application of the existing diagnostic frameworks ICD-10 (International Classification of Diseases, Tenth Revision) and DSM-V (Diagnostic and Statistical Manual of Mental Disorders, Fifth Edition) by specialists. What is more probable is that diagnoses are entirely missing because no systematic screening is taking place, but a specialist is only called at the internal medicine department physician’s suspicion. Diagnoses from the medical history of the patient are also not included but could have included mental illness. That is why this article is aiming to recommend early consideration of possible mental illness diagnoses by specialists. We added a note in this regard to the discussion.

3) The statistical analysis seems to me to have room for improvement. Since the data were collected across different departments with probably different mean lengths of stay, I think a multi-level model with the departments as clusters could be considered. If varying slopes were allowed, the effects could be estimated in a more differentiated way.

Yes, the data were collected across six different departments. As suggested in the next comment, the varying mean lengths of stay were added in S7 Table. We agree that the detectable variation could be represented more accurately in a multi-level model. However, we decided on the current model architecture for the following reasons: (i) The variable hospital department and main diagnosis correlate strongly (Cramer’s V=0.52), (ii) the aim for this study was a simple, intuitive model to serve as a benchmark for future studies. We nevertheless added this idea to the limitations in the discussion. Please also refer to the new S4 Supplemental Information for more details on the model selection process. 

4) It would be helpful for the reader to have more information on the departments involved. Could the lengths of stay and the proportions of cases with comorbidities be broken down by department?

We added the supplemental S7 Table that shows the average length-of-stay and proportions of cases with mental comorbidity broken down by the 6 hospital departments that were part of this data set.

---

## [Decision Letter · Decision Letter 1]

2 Jun 2023

­Interaction of mental comorbidity and physical multimorbidity predicts length-of-stay in medical inpatients

PONE-D-23-03643R1

Dear Dr. Stahl-Toyota,

We’re pleased to inform you that your manuscript has been judged scientifically suitable for publication and will be formally accepted for publication once it meets all outstanding technical requirements.

Kind regards,

Sebastien Kenmoe

Academic Editor

PLOS ONE

Additional Editor Comments (optional):

Reviewers' comments:

Reviewer's Responses to Questions

**Comments to the Author**

1. If the authors have adequately addressed your comments raised in a previous round of review and you feel that this manuscript is now acceptable for publication, you may indicate that here to bypass the “Comments to the Author” section, enter your conflict of interest statement in the “Confidential to Editor” section, and submit your "Accept" recommendation.

Reviewer #1: (No Response)

Reviewer #2: All comments have been addressed

Reviewer #3: All comments have been addressed

2. Is the manuscript technically sound, and do the data support the conclusions?

Reviewer #1: Partly

Reviewer #2: Yes

Reviewer #3: Yes

3. Has the statistical analysis been performed appropriately and rigorously? 

Reviewer #1: No

Reviewer #2: Yes

Reviewer #3: Yes

4. Have the authors made all data underlying the findings in their manuscript fully available?

Reviewer #1: No

Reviewer #2: (No Response)

Reviewer #3: No

5. Is the manuscript presented in an intelligible fashion and written in standard English?

Reviewer #1: Yes

Reviewer #2: (No Response)

Reviewer #3: Yes

6. Review Comments to the Author

Reviewer #1: Th authors adequately responded on several issues that were pointed out in the text, but for some important topics readers are referred to supplementary material and future studies. This makes the review process something of a quest for information. Here results are mostly expressed as means and SD, whereas the negative binomial models implicate that these are not very helpful descriptive measures. Minor note: in S4_Supporting_information (page 6) Glm function in R is referred to as general linear model, whereas all models compared are generalized linear model with different distribution families.

Furthermore, the authors show a very pragmatic perspective when reporting statistical analyses: “the aim for this study was a simple, intuitive model to serve as a benchmark for future studies”. But the model presented is not that simple or intuitive. As the variable hospital department and main diagnosis correlated strongly, it is unclear how to distinguish between department-effect and diagnosis-effect. LOS-estimates based on the model are presented as point estimates instead of expected LOS-values with prediction intervals.

In addition, interpretation of the interaction effect is still confusing. The authors conclude that “our study provides evidence … that mental comorbidity predicts length-of-stay dependent on physical multimorbidity.” But this is reversed: for low Elixhauser scores there is no difference in LOS, as this score increase the LOS increases, but in the group with mental comorbidity we see that higher Elixhauser have a stronger impact on LOS. Or as the authors rephrase it: “The interaction between mental comorbidity presence and Elixhauser score can be interpreted in the sense that for patients with mental comorbidity, the effect of increasing levels of multimorbidity on length-of-stay was stronger.” But this is descriptive and not really an interpretation of the interaction effect. Supplementary material 1 shows that the largest differences in median LOS-values between Yes/No mental comorbidity are found for ‘Neoplasms’ and ‘Injury, poisoning and certain other consequences of external causes’. Ideas are lacking on why these associations were found and how they could be linked to physical multimorbidity.

It is up to the editors to decide if this ‘benchmark’ for future studies is suitable for publication. In my view this kind of theory-poor applied research only adds to the body of not replicable findings.

Reviewer #2: The authors addressed all queries and amended the manuscript according to the comments. I have no further comments

Reviewer #3: Dear Authors,

my comments have been duly taken into account and I support the publication of the manuscript.

With best regards

7. PLOS authors have the option to publish the peer review history of their article (what does this mean?). If published, this will include your full peer review and any attached files.

Reviewer #1: **Yes: **André Wierdsma

Reviewer #2: No

Reviewer #3: No

---

## [Editor Report · Acceptance letter]

6 Jun 2023

PONE-D-23-03643R1 

Interaction of mental comorbidity and physical multimorbidity predicts length-of-stay in medical inpatients 

Dear Dr. Stahl-Toyota:

I'm pleased to inform you that your manuscript has been deemed suitable for publication in PLOS ONE. Congratulations! Your manuscript is now with our production department. 

Kind regards, 

on behalf of

Dr. Sebastien Kenmoe 

Academic Editor

PLOS ONE